# Shaping in the Third Direction: Colloidal Photonic Crystals with Quadratic Surfaces Self-Assembled by Hanging-Drop Method

**DOI:** 10.3390/polym16131931

**Published:** 2024-07-06

**Authors:** Ion Sandu, Iulia Antohe, Claudiu Teodor Fleaca, Florian Dumitrache, Iuliana Urzica, Marius Dumitru

**Affiliations:** 1Lasers Department, National Institute for Lasers, Plasma and Radiation Physics, 409 Atomistilor Street, 077125 Magurele, Romania; ion.sandu@inflpr.ro (I.S.); iulia.antohe@inflpr.ro (I.A.); claudiu.fleaca@inflpr.ro (C.T.F.); florian.dumitrache@inflro.ro (F.D.); iuliana.iordache@inflpr.ro (I.U.); 2Romanian Academy of Scientists (AOSR), 54 Splaiul Independentei, 050094 Bucharest, Romania

**Keywords:** colloidal photonic crystals, self-assembly, quadratic surfaces, metasurfaces

## Abstract

High-quality, 3D-shaped, SiO_2_ colloidal photonic crystals (ellipsoids, hyperboloids, and others) were fabricated by self-assembly. They possess a quadratic surface and are wide-angle-independent, direction-dependent, diffractive reflection crystals. Their size varies between 1 and 5 mm and can be achieved as mechanical-resistant, free-standing, thick (hundreds of ordered layers) objects. High-quality, 3D-shaped, polystyrene inverse-opal photonic superstructures (highly similar to diatom frustules) were synthesized by using an inside infiltration method as wide-angle-independent, reflective diffraction objects. They possess multiple reflection bands given by their special architecture (a torus on the top of an ellipsoid) and by their different sized holes (384 nm and 264 nm). Our hanging-drop self-assembly approach uses setups which deform the shape of an ordinary spherical drop; thus, the colloidal self-assembly takes place on a non-axisymmetric liquid/air interface. The deformed drop surface is a kind of topological interface which changes its shape in time, remaining as a quality template for the self-assembly process. Three-dimensional-shaped colloidal photonic crystals might be used as devices for future spectrophotometers, aspheric or freeform diffracting mirrors, or metasurfaces for experiments regarding space-time curvature analogy.

## 1. Introduction

Colloidal crystals (CCs) consist of ordered arrays of submicron-sized colloidal particles, typically silica or polystyrene hard spheres. These crystals self-assemble from a liquid medium using bottom-up techniques or in combination with top-down technologies [1,2,3,4,5,6]. Their defining characteristics include long-distance order and porosity. The pores within CCs allow for the infiltration of various fluids, including gases, molecular/ionic solutions, and even smaller colloids. As these fluids undergo phase changes (gas or liquid to solid) followed by the dissolution of spheres (via calcination or chemical processes), a unique type of crystal known as inverted opals (IOs) emerges [2,3,4,5,6,7]. Colloidal crystals share a close relationship with photonic crystals (PhCs) [8,9,10,11]. When the size of colloidal particles matches the wavelength of the electromagnetic waves they interact with, both types of crystals exhibit the same key concept: the photonic band gap (PBG) [12,13], which represents a range of frequencies that are forbidden to exist within the interior of the crystals. This phenomenon gives rise to various optical effects, including Bragg diffraction, fluorescence suppression, electroluminescence, and lasing [14,15,16], that enable band gap materials to operate, confine, and control the propagation of light within extremely small volumes. Consequently, they find applications in the fabrication of resonant, tunneling, wave-guiding, and filtering optical devices. Examples include switches, memories, light sources, low- and high-reflection coatings, lenses, mirrors, laser components, and sensors, addressing the growing demand for sensing devices in healthcare and physical parameter monitoring, defense, food quality control, and aerospace [17,18,19,20,21].

When the size of colloidal spheres falls within the visible light domain, colloidal crystals (CCs) and photonic crystals (PhCs) [22,23,24], but also non-gap materials such as “diffractive gratings” [25,26] or certain living organisms [27,28], exhibit a fascinating phenomenon known as “structural colors”. In terms of structural and morphological complexity, these natural organisms remain unparalleled when compared to photonic devices crafted by human hands. They boast 3D-shaped photonic structures—concavities, convexities, and even free-form designs—demonstrating precise control over structuring in all three dimensions. The typical outcome of colloidal self-assembly is a single layer, a bi-layer, or a thin film composed of closely packed spheres. These spheres form either a hexagonal close-packed (hcp) or a face-centered cubic (fcc) lattice, representing the simplest structure with a PBG [14,16]. These structures resemble straightforward “bricks”, devoid of any special macroscopic shapes, where geometric optics becomes irrelevant relative to wave optics. While most colloidal crystals adhere to this basic form, there have been rare attempts at more intricate designs (prisms for example [29,30]). Curved colloidal crystals—such as spheres, hemispheres, cylinders, doughnut-like shapes, or tori—have emerged through self-assembly processes [31,32,33,34,35,36,37,38,39,40]. Alternative approaches, such as introducing buckling into an originally planar structural layout [41,42] or utilizing stretchable membranes [43,44], have also yielded interesting results. Despite successful molecular self-assembly in constructing 3D-shaped molecular crystals within the ultraviolet wavelength domain [45], and more recently, the production of shaped colloidal crystals (resembling infiltrated opals) in the visible light size range through 3D printing [46,47], these achievements fall short of our expectations for such a fundamental phenomenon as self-assembly. Complex optical phenomena remain unexplored, and a deeper understanding of how light pathways depend on the shape of the medium it traverses could lead to the creation of intriguing devices. However, we have already gleaned valuable insights from the study of spherical structures—often referred to as super-balls or super-spheres [48,49]. Here are some shape-dependent optical properties we have discovered by reading the literature:(a)Show constant colors at different viewing angles (angular independence).(b)Both Bragg and grating diffraction contribute to the reflection.(c)Transmission spectra are not the inverse of the reflection.(d)Different crystal planes may contribute to the reflection.

Our understanding of shape-dependent optical properties remains limited when it comes to cylindrical colloidal crystals. These crystals exhibit axial symmetry due to their cylindrical geometry. When incident light is perpendicular to the fiber axis, the periodic-structured fiber displays non-iridescent color. However, iridescence emerges if there is a nonzero axial component in the incident light direction. The structural colors of these fibers result from two phenomena: reflection due to the photonic band gap and Mie scattering of the colloidal spheres [50].

Interestingly, the literature lacks reports of shape-dependent phenomena for torus or doughnut-like colloidal crystals. However, we have encountered a small number of suggested applications related to shape dependence different from that of wavelength angular independence [39,40]. These applications leverage simultaneous Bragg and grating diffraction phenomena in convex colloidal crystals.

By carefully arranging shaped colloidal crystals, we can create patterned films. For instance: spherical CCs composed films (bead films) which exhibit structural colors similar to their spherical CC elements and maintain consistent colors across different viewing angles or novel photonic paper where, by using PEG hydrogel to replicate spherical CC-composed films, some authors achieved a flexible, vividly colored photonic paper with wide viewing angles and rewritable surfaces. This paper holds promise for potential information displays [51]. As for cylindrical CC films, no new shape-dependent phenomenon has emerged. This leaves us wondering about the appearance of CC films beyond spheres and cylinders—perhaps ellipsoids or other intriguing shapes.

Aspherical and freeform optical components have already demonstrated remarkable qualities [52,53]. Their surfaces can correct system aberrations, enhance the field of view, and improve numerical aperture. Consequently, image quality in optical imaging systems is significantly enhanced. Now, let us explore how these principles apply to colloidal photonic crystals. These crystals would greatly benefit from their surface shapes and local variations in inclination. By incorporating these features, their degree of freedom increases, leading to improvements in their photonic band gap properties. Imagine colloidal crystals shaped as ellipsoids, paraboloids, or hyperboloids—objects with quadratic surfaces. These shapes could play a crucial role in experiments where 2D photonic crystals are curved in 3D space to study space-time curvature analogy [54,55,56,57,58], where the underlying concept is transformation optics based on metasurfaces [59]. Additionally, these same curved colloidal crystals could be employed to investigate absolute optical instruments. These instruments stigmatically bring an infinite number of light rays from a source to its image. Furthermore, they offer insights into the paths of light rays and waves on geodesic lenses [60].

Producing curved colloidal crystals is our initial challenge, and self-assembly emerges as a promising approach. The aim of this work is to find the experimental conditions in which colloidal particles self-assemble as quality colloidal photonic crystals, other than spherical ones, which were obtained in ref. [39]. We report as a novelty the synthesis of freeform colloidal photonic crystals by the self-assembly method and some of their shape-dependent optical phenomena.

## 2. Materials and Methods

### 2.1. Materials

We used SiO_2_ submicron spheres with mean diameters of 0.264 µm and 0.384 µm (at 5% *w*/*v*) and polystyrene (PS) spheres with a size of 20.00 μm (at 10% *w*/*v*) as aqueous colloidal solutions. These materials were directly sourced from microParticles GmbH, Volmerstr. 9A, H3.5.1, D-12489 Berlin, GERMANY For the self-assembly of colloidal crystals, we employed various substrates:(a)Smooth and clean glass fibers: These common glass fibers were 5 cm long and ranged from 0.2 mm to 3 mm in diameter.(b)Microscope glass slides.(c)Copper and steel wires: These wires were also 5 cm long and had diameters ranging from 0.2 mm to 3 mm.(d)Steel spheres: These spheres were 5 mm in diameter.(e)PET (polyethyleneterephtalate) concavities: We obtained these concavities (with a diameter of 4 mm and a depth of 1.5 mm) from the edge of the packaging of a KD-JECT^®^ III 1 mL syringe, KD Medical GMBH Hospital, Charlottenstrasse 65, 10117 Berlin, Germany.

To deposit the colloidal solution onto the substrates, we used standard and commercially available KD-JECT^®^ III 1 mL syringes (incorporating a G 29 needle).

### 2.2. Methods

#### 2.2.1. Opal-Like Colloidal Photonic Crystal Self-Assembly

(a)Opal-like colloidal photonic crystal on a fiber: We began by placing a glass or metallic fiber substrate either horizontally or at a 30–60° tilt angle using a small metallic burette clamp. Next, a droplet of 0.264 µm of SiO_2_ colloidal solution was carefully formed at the tip of a syringe. The droplet was gently transferred to the fiber: For horizontal substrates, it was touched somewhere along the fiber’s length. For tilted substrates, it was touched at the end of the fiber. If a larger drop volume was needed, additional droplets were added to the existing one hanging on the fiber. The droplet was allowed to dry under normal laboratory conditions, typically taking 30–60 min.(b)Opal-like colloidal photonic crystal on a metallic coil spring: A copper wire (0.3 mm in diameter) was wound around a fiber (2 mm in diameter) to create a coil spring with approximately 10 loops and a 1 mm distance between loops. The detached coil spring was then placed horizontally. A hanging 0.264 µm SiO_2_ colloidal drop was attached to the metallic coil spring, starting from its inside and gradually increasing the drop volume by adding more droplets. Finally, the drop was allowed to dry.(c)Opal-like colloidal photonic crystal on tangent metallic spheres: Two touching metallic spheres (each 5 mm in diameter) were fixed in a polystyrene thick film on a microscope glass slide (achieved by melting polystyrene flakes onto the glass slide). The glass slide was inverted and placed horizontally. A hanging 0.264 µm SiO_2_ colloidal drop was formed and attached to the spheres, hanging between them, and left to dry.(d)Opal-like colloidal photonic crystal self-assembled on a polymeric concavity: The PET sheet of the KD-JECT^®^ III syringe packaging was placed in a horizontal position, its concavity pointing downwards. A hanging 0.264 µm SiO_2_ colloidal drop was formed and attached to the concavity and left to dry.

#### 2.2.2. Polystyrene Inverse-Opal Superstructure Fabrication

(a)Ellipsoid inverse-opal superstructure fabrication: A polystyrene (20.00 µm spheres) opal-like colloidal crystal has self-assembled on a horizontal fiber as in 2.2.1 (a). After drying, by keeping it in the same hanging position, a 0.264 µm SiO_2_ colloidal solution drop was gently transferred to its top. The SiO_2_ colloidal solution infiltrates and crystallizes between the polystyrene spheres. After polystyrene melting and infiltration (270 °C, 15 min), polystyrene solidification, SiO_2_ dissolution (25% HF), and water washing, a high-quality polystyrene inverse opal resulted.(b)Torus onto ellipsoid inverse-opal superstructure fabrication: A polystyrene (20.00 µm spheres) opal-like colloidal photonic crystal has self-assembled on a fiber as in 2.2.1 (a). After drying, by keeping it in the same hanging position, a 0.264 µm SiO_2_ colloidal solution drop was gently transferred to its top. The SiO_2_ colloidal solution infiltrates and crystallizes between the polystyrene spheres. After drying, a second 0.264 µm SiO_2_ or 0.384 µm SiO_2_ colloidal solution drop was gently transferred to its top. The second drop cannot infiltrate (all former holes between PS spheres are already filled with silica spheres) and forms a toroidal-shaped deposit on the ellipsoid surface. Melting infiltration and casting, followed by HF dissolution, give rise to an inverse-opal polystyrene colloidal photonic crystal of a special architecture.

### 2.3. Reproducibility

The hanging-drop method is a highly reproducible one. The specific shape of colloidal crystals is obtained each time within the limits of experimental parameters. The high quality of sphere ordering is not influenced by synthesis parameters such as relative humidity, temperature (up to 80 °C), concentration (up to 10 wt.%), external vibrations and shocks, or their short transitory variations. However, as with any other self-assembly method, it is highly sensitive to size dispersion of the spheres from colloidal solution.

### 2.4. Investigations

Macro-scale observations of the as-synthesized patterned colloidal crystals were performed using a zoom camera and optical microscopy. A scanning electron microscope (SEM) (Apreo S Thermo Fisher Scientific, Auburn, AL, USA) was used to observe the structures and morphologies of the self-assembled patterned colloidal crystals. UV–Vis reflectance spectra were acquired through a fiber optic system by using an AvaLight-DHc light source and an AvaSpec-ULS2048CL-EVO high-resolution spectrometer, both from Avantes, Apeldoorn, The Netherlands. The fiber-optic probe has a special configuration where six optical fibers which serve as light white sources surround a seventh one which is used to receive the reflected light.

## 3. Results and Discussion

### 3.1. Synthesis and Investigation of Quadratic Surface Colloidal Crystals

Recently, we employed the hanging-drop method [61] to fabricate convex colloidal photonic crystals resembling the cap of a sphere [39]. Our findings revealed two crucial factors that enhance self-assembly in this configuration: (a)The tangential component of the weight of colloidal spheres (denoted as G_T_) confines the spheres within a compact colloidal crystal (Figure 1a).(b)The normal component of gravity (G_N_) generates a zero static frictional force (F_F_) between the spheres and the liquid/air interface. This allows the entire system to continuously reconfigure itself at the microscopic level (until it “freezes”), effectively creating a defect-free substrate (see Figure 1a).

The shape that colloidal crystals adopt through the hanging-drop method depends on various factors, including drop volume and the substrate’s dimensional extension (0D or 2D). Let us explore these scenarios: When using a 2D substrate, hundreds of microliters of colloid can form a substantial hanging drop (Figure 1b). After solution evaporation, this results in a large, bump-like colloidal crystal. In contrast, with a 0D substrate (such as the tip of a vertically positioned fiber), only microliters of colloid can hang (Figure 1b).

As the drop dries (Figure 1b, case 2D and case 0D [39]), it forms a spherical colloidal crystal. If the drop hangs on a 2D substrate, it always forms a spheroidal colloidal crystal. If a drop hangs on a 0D substrate, it always forms a sphere-cap colloidal crystal. In both cases, the ordering distance (the size of a perfect single domain) remains sufficiently large and does not depend on the initial drop size. This leads us to believe that the self-assembly occurring at the liquid/air interface of the hanging droplet could be also independent of the droplet’s shape as its lower part remains contact-free. Our recent findings suggest that achieving non-spherical colloidal crystals is feasible by harnessing three fundamental forces: cohesive force (F_C_) between solvent molecules, adhesion force (F_A_) between the colloidal solution and the substrate, and gravity (G). These forces work in concert within specific experimental setups, allowing us to manipulate colloidal crystal shapes effectively. What if we could modify the drop shape—bending or stretching it by using a 1D substrate as depicted in Figure 1b—while simultaneously ensuring that the liquid/air interface at the lower part remains contact-free? Would this lead to the self-assembly of an elongated colloidal crystal?

If a colloidal drop touches a glass, a metallic, or even a polymeric fiber, it adheres to the fiber surface due to the adhesion force. Depending on the drop volume, contact angle, fiber nature, roughness, diameter, and tilting angle, the drop partially spreads and takes different static shapes. It takes a spheroidal shape (for drops above a certain volume) if it hangs on a vertically placed fiber tip [39] and an axisymmetric barrel or an asymmetric (with respect to the fiber axis) clam shell if the drop is deposited on a horizontal fiber [62,63]. The drops take a clam shell shape for volumes larger than a certain value because of the influence of gravity (heavy drops). They are elongated along the fiber (Figure 2a) and compressed across the fiber (Figure 2b).

Using a large volume (tens of microliters, 3–5 mm in size) of 264 nm SiO_2_ colloidal solution (Figure 2a,b), we created a localized deposit (several millimeters long) on the fiber surface at the end of evaporation process (Figure 2c). When ambient light reflects off this deposit, it reveals intense colored spots. However, when observed with the naked eye, the colors appear somewhat uneven.

The deposit takes the form of a 3D curved ellipsoidal ring, with a 3D ellipsoidal colloidal crystal positioned at the center of the ring. Most of the silica nanospheres are concentrated either in the ring or within the ellipsoidal colloidal crystal. The most striking reflected color is light green, which appears to be independent of the viewing angle. However, when observed from different zones at the same angle, other colors besides green become visible. While the formation of a ring could be anticipated based on previous work by other authors (involving small and sessile droplets on fibers) [64,65], the ellipsoidal shape of the deposit from the center and the high-quality self-assembled structure (ring and ellipsoid) in our case were surprising. By using as substrates fibers of different nature (glass, metal, and plastic) which present different contact angle at their triple-contact line with the colloidal drop, we observed that materials with a lower contact angle such as glass/water support a higher volume of colloid against gravity than metal or plastic. This leads to thicker and more mechanical-resistant colloidal crystals. By mixing ethanol with colloidal solution in different ratios, the adherence force between colloid and fiber increases, elongating the drops along the fiber. However, this leads also to decreases in cohesion force; thus, a much lower volume can be suspended and subsequently resulting in thinner and more elongated colloidal crystals.

The long-range order of self-assembled SiO_2_ spheres in both the ring and ellipsoid structures was confirmed through UV–Vis retro-reflection using a fiber-optic system (Figure 2c). By examining reflection spectra from three different zones of the colloidal structure, we observed two distinct reflection peaks in each spectrum:

**Peak λ_1_**: Located at either 535 nm or 555 nm. This peak corresponds to reflection from (111) planes in a face-centered cubic (fcc) lattice, as determined by Bragg diffraction. The refractive index (n) used for calculations was 1.42, considering 264 nm diameter silica spheres in a retro-reflection configuration [16].

**Peak λ_2_**: Found around half the value given by (111) family planes (approximately 276 nm or 298 nm). This peak typically results from the superposition of (200), (220), (311), and (222) family planes [16]. At a certain Bragg reflection angle, this superposed peak begins to decompose into its individual components [16].

Interestingly, the 20 nm shifts observed in both reflection bands appear to correlate with the local tilting of each zone. The high quality of the self-assembled structure (ring and ellipsoid) is evident from their narrow Full Width at Half Maximum (FWHM) of approximately 7%. 

Optical microscopy images of the ellipsoidal crystal (Figure 2d) reveal magnificent blazing colors: yellow at the image center, gradually transitioning radially to light green, light blue, and dark blue. The yellow color likely arises from Bragg diffraction off (111) family planes in the fcc lattice. The iridescence phenomenon (green-to-blue transition) is attributed to Bragg reflection on a convex surface (mirror), which presents a higher field of view and a higher angular field of view than a planar one. Bragg-diffracted light at large incidence angles will be reflected inside the acceptance cone of the microscope objective, while in the case of a plane mirror, it will fall outside. 

When a colloidal drop hangs on a tilted fiber (at an angle of 30–60°), the resulting crystalline structure becomes even more captivating, quite different from both those resulting when a drop is placed on a vertical fiber tip [39] or when the drop hangs on a horizontal fiber (Figure 2c). It closely resembles the “Inner-Loop Limançon” curve [66], also known as the Pascal snail (Figure 2f). Notably, neither UV–Vis spectroscopy (inset in Figure 2g) nor optical microscopy images (Figure 2g,h) reveal significant differences between structures self-assembled on horizontal fibers (Figure 2c) versus those on tilted fibers (Figure 2g,h). However, there is one intriguing detail: the height of the peak resulting from superposed planes (at 298 nm) is greater than that from (111) planes (at 555 nm) for ellipsoids self-assembled on tilted fibers compared to those on horizontal ones. This difference may arise from their slightly different curvatures. Additionally, we must draw attention to the blue light spot on the side of the ellipsoids in Figure 2h. Remarkably, this spot is not an effect of diffraction or reflection from the microscope beam light falling normally on the sample. Its cause will be explored further in this article.

In scanning electron microscopy (SEM) investigations, we examined self-assembled colloidal crystals formed by larger-sized (20 μm) polystyrene colloidal spheres (Figure 3a–c) and by 0.264 μm silica spheres (Figure 3d–f) adhering to horizontal fibers.

We observed that the self-assembly phenomenon is independent of sphere nature and size. Both types of colloidal crystals exhibited high-quality surfaces. Notably, we observed no cracks at either the micro- or macroscopic level (except those caused by handling errors). However, upon closer examination of a handling-induced crack (Figure 3d), we confirmed higher ordering of the colloidal crystal on its cross-section, particularly near its surface (Figure 3g). Unfortunately, despite the good quality of the external surface of the colloidal crystals, their interface with the fiber exhibited poor optical quality and UV–Vis spectroscopy response. Interestingly, this seems to be a common characteristic of self-assembled colloidal crystals using the hanging-drop method. However, in a recent article, we demonstrated how the quality of this interface could be improved when intentionally seeking higher quality [39].

We conducted experimental investigations involving colloidal drops hanging on multiple fibers or curved in various configurations (Figure 4a–d). The resulting structures appear rather trivial: ellipsoids, spheroids, or indistinct shapes. However, despite their lack of clear form, the external surface quality remains high (Figure 4e).

Now, let us delve into more intriguing configurations: (a)Colloidal drop on a horizontal metallic coil spring (Figure 4f): This setup produces connected hyperboloids (Figure 4g) which exhibit an unexpected reflection of ambient light. The green rectangular spots represent light reflected from an ordinary white light source mounted on the ceiling, positioned three meters above the sample surface.(b)Colloidal drop on two tangent metallic spheres (Figure 4h): A similar yet unique hyperboloid emerges in this case. Interestingly, although both structures are hyperboloids, they reflect light slightly differently (Figure 4i).(c)Colloidal drop hanging inside a macroscopic concavity (Figure 4j): Consider the scenario depicted in Figure 4j (left): a colloidal drop deposited and hanging within a macroscopic concavity (a kind of freeform surface). Surprisingly, as we increased the drop volume beyond the concavity’s capacity, the liquid did not spill over the edge as expected. Instead, it descended, forming a massive drop (Figure 4j (right)). The interplay of adhesion, cohesion, and gravity balanced in an unexpected manner, yet meeting the necessary conditions for high-quality colloidal crystal formation: a large colloid volume acting as a reservoir, and the drop’s bottom surface free from contact.

Remarkably, as the colloidal drop evaporated in this configuration, it self-assembled into an extensive, well-ordered colloidal crystal (Figure 4k). The resulting shape resembles that of a cylindrical shell, with one side closely following the substrate’s curvature, while the other face presents a second curvature akin to the inner part of a torus. The convex side exhibits wide-angle independence, reflecting and diffracting white light with high intensity (Figure 4k).

If we consider the structures presented earlier (such as the hemisphere [39] and the bump-like [61]), a common characteristic emerges among all colloidal crystals self-assembled using the hanging-drop method. They belong to a class of 3D objects with quadratic surfaces. These surfaces are known as “quadric surfaces”—algebraic surfaces defined by quadratic (order 2) polynomials. In 3D Euclidean space, they take the forms of ellipsoids, paraboloids, or hyperboloids. In the field of optics, most optical devices are either spherical or aspheric. Spherical lenses often suffer from spherical aberration (resulting in blurry images), while aspheric lenses focus light to a single point or correct image distortions like astigmatism. Large aspheric lenses find applications in telescopes and cameras, while smaller aspherical lenses play essential roles in fiber-optic networks, laser devices, and surgical equipment. However, the use of spherical, aspheric, or freeform diffractive devices is less common, and the simultaneous study of wave and geometric optical phenomena in such structures remains limited. A freeform colloidal crystal was produced for the first time by the direct-writing technique [67]. However, the hanging-drop method seems to be also able to fabricate such free-forms that contain quadratic surfaces. The hanging-drop method is weakly dependent on relative humidity in the air, rate of evaporation, temperature, and concentration. However, the colloidal crystal shape and its mechanical resistance are assured by a large amount of solid spheres, which are usually provided by a high colloid concentration, while their organization quality by a limited concentration (up to 10% wt.). These opposite requirements can be accommodated by using large volumes of initial colloidal solutions in contact with the substrates. This means a limited range of substrates which can assure a high adherence force or/and an increased contact surface between substrates and liquid colloids.

In the following sections, we will present results obtained from investigating colloidal crystals with quadratic surfaces, exploring their potential shape-dependent phenomena.

### 3.2. Optical Phenomena in Quadratic Surface Colloidal Crystals

A flat colloidal crystal self-assembled by capillarity between two parallel glass slides using a similar setup as in [68] and a set of quadratic surface colloidal crystals were produced by the hanging-drop method (Figure 5a–e) or by the “edge effect” (Figure 5f) [38].

A set of retro-reflection UV–Vis spectroscopy measurements were performed on each of these crystals (excepting the hyperboloid produced between two metallic spheres) by varying the distance between their surface and the probe. The resulted spectra are shown in Figure 6. 

Analyzing these spectra, we made several observations:(a)The reflectance band corresponding to (111) planes increases up to a certain point (at 2.5 mm), followed by a decrease as the distance increases.(b)At a specific distance between the sample and the probe, the reflection band associated with (200) planes appears for the curved colloidal crystals but not for the flat ones. Its intensity increases with the increasing distance from the probe for all crystal shapes.(c)The (111) peak position exhibits a small blue shift at greater distances from the probe, with the shift size increasing as the crystal curvature becomes more pronounced

To correctly interpret these observations, we must consider that the probe is not a simple optical fiber; it forms a complex optical system. This system comprises six optical fibers serving as light white sources, surrounding a seventh fiber used to receive reflected light. Additionally, a focusing lens is positioned in front of all these fibers.

*Observation (a)*: The reflected light intensity follows the distance to the lens focus.

*Observation (b)*: We need to account for various probe fabrication parameters, including: acceptance angle, focal distance, focal spot diameter, working distance, front of view, and angular front of view. Typically, angular front of view and focal distance are fixed parameters optimized for specific working distances on flat samples. To increase the angular front of view, we simply moved the sample farther from the probe during our experiments. This maneuver allowed a portion of the incident light beam to intercept the crystal’s (200) family planes from the proper angle, ensuring their reflection remained within the acceptance angle of the central optical fiber. While this adjustment did not significantly impact flat samples, it became relevant for convex-shaped or prismatic colloidal crystals. Additionally, moving the sample away from the focus reduced the light intensity falling on the (111) family planes. Calculating the angle between the incident light beam and the surface normal at specific points on a three-axis (scalene) ellipsoid colloidal crystal is complex. Differences in (200) intensity variations for different shapes may be induced by varying curvatures.

*Observation (c)*: The small blue shift in the (111) peak position could be linked to the decrease in reflected light intensity as the distance to the probe increases [69]. Unfortunately, the scientific literature provides limited information on this phenomenon. Nevertheless, despite its subtle and fuzzy nature, this phenomenon could serve as an interesting approach for revealing certain optical properties of shaped colloidal crystals.

Next, let us discuss the phenomenon of blue light occasionally observed on the sides of ellipsoidal colloidal crystals (but not exclusively) from Figure 2h. This phenomenon has been highly intriguing because it appears to be unpredictable and unreproducible. It manifests independently of how well we “control” the experimental setup. However, the explanation is surprisingly straightforward: an occasionally open window shutter allows diffuse light from outside to fall onto the sample as a secondary light source as schematically presented in Figure 7a. We can observe this phenomenon in Figure 7b–d, which depicts three distinct situations: both light sources, microscope light only, and diffuse sunlight (coming in through the window) only.

This phenomenon reveals a very important characteristic of shaped colloidal crystals: their reflection is directionally dependent. We can observe this effect in Figure 7e–h, where the same ellipsoidal colloidal crystal is rotated in-plane, alternately illuminating its sides. Interestingly, we notice that the side light, diffracted and reflected from the surfaces along the ellipsoid’s main axis, reaches the microscope objective. However, this effect does not occur along the ellipsoid’s short axis and does not influence the normal light from the microscope, which remains unchanged.

The light blue color corresponds to the light wavelength that Bragg diffracts from the (111) family planes at an incident angle of around 45°. This implies that only the light falling from the ‘horizon’ of the ellipsoid can produce the blue light. However, in Bragg diffraction, the reflection angle must equal the incident angle relative to the normal to the surface. While determining the normal to the surface is straightforward for flat objects, it becomes more complicated for curved ones. Thus, theoretically determining the exact angle between the incident beam and the reflected one that causes the side blue light on ellipsoidal (but not exclusively) colloidal crystals requires knowledge of the local tilting of the surface. 

In a series of consecutive experiments (Figure 7i–l), we kept an ellipsoidal colloidal crystal stationary. The optical microscope illuminated the upper surface of the crystal with normal light. Simultaneously, we adjusted the maximum angle at which light from the secondary source (the diffuse sunlight coming in through the window) fell on the crystal’s side surface—a kind of solid angle. By raising the shutter, we increased the light flow. Our observations revealed that the side-reflected light underwent a subtle change in wavelength, transitioning from dark blue to light blue. Additionally, the ellipsoid’s surface area which reflects the secondary light source increased proportionally with the illumination angle. Interestingly, beyond a certain light flux from the secondary source, the wavelength of diffracted/reflected light observed through the microscope (yellow-green) was replaced by the wavelength of side diffraction/reflection (light blue), all while maintaining the geometrical shape of the microscope’s focusing spot (slightly ellipsoidal). This captivating optical phenomenon necessitates consideration of both directional dependence and light source intensity.

### 3.3. Synthesis and Investigation of Shaped Super-Structured Inverted Opals

The shaped colloidal crystals synthesized through the hanging-drop method exhibit intriguing mechanical and wetting properties, rendering them valuable as templates for creating even more sophisticated devices. These crystals consist of hundreds of ordered layers, providing robust mechanical resistance against future manipulations such as rewetting. Remarkably, they can be readily infiltrated with water-based solutions without disintegrating or detaching from the substrate. In a previous study [70], we constructed a 2D superstructure by arranging a single layer of large-sized (50 μm) polystyrene microspheres. These spheres were infiltrated with even smaller silica (264 nm) spheres, leading to their self-assembly through solvent evaporation. Subsequently, we achieved SiO_2_ crystalline structure by melt infiltration of polystyrene from the inside, followed by the dissolution of the silica spheres. In this work, we explore the fabrication of shaped superstructures using a similar approach. Specifically, we demonstrate the production of two distinct shapes: a sphere (resembling the cap of a sphere) and an ellipsoid. Our process begins by depositing 20 μm polystyrene colloidal spheres as a hanging drop at the tip of a vertically positioned fiber in one case and on a horizontal fiber in the other (as depicted in Figure 8a).

At the end of the drying process, both spherical and ellipsoidal colloidal crystals formed on the fibers. These crystals remained in their original positions as a 264 nm SiO_2_ colloidal droplet was gently deposited onto the top of each polystyrene crystal. The droplet infiltrated the crystal pores and self-assembled into a robust structure after drying (Figure 8b). This structure exhibited mechanical resistance to the polystyrene melt infiltration caused by the heated PS spheres within the entire system. Subsequent thermal treatment, PS solidification, and silica HF dissolution (Figure 8c) resulted in the formation of inverse opal superstructures with spherical and ellipsoidal shapes (as depicted in Figure 8d,e). Although their external morphologies differed, their internal structures were similar, characterized by highly ordered 264 nm interconnected holes throughout the volume. Notably, the initial 20 μm spheres failed to transform into interconnected 20 μm holes due to an excess of polystyrene material. However, each 20 μm sphere from the surface layer exhibited a slightly concave shape.

The UV–Vis spectra of the two differently shaped structures exhibit striking similarities (as shown in Figure 8h,i). These spectra reveal a prominent and narrow reflection band centered around 485 nm, along with a secondary, lower band at 290 nm. Remarkably, both reflection bands closely align with calculated values based on Bragg’s law, considering a polystyrene refractive index of n = 1.59 and a hole size of 264 nm (corresponding to (111) family planes and superposition of other reflection bands, respectively).

Notably, optical microscopy images reveal intriguing features: a light blue spot at the center (Figure 8h,i) and a surrounding dark blue halo—a limited iridescence. This halo arises from the high-quality external surface layer; thus, it becomes possible to fabricate either spherical or aspherical convex reflective diffraction colloidal crystals. These shaped superstructures hold promise for applications in metastructures, metamaterials, and metasurfaces [71,72,73]. Furthermore, since these structures consist solely of transparent polystyrene material within the visible wavelength range, they hold theoretical potential for use as convex diffractive lenses, expanding their practical applications.

When a second 264 nm SiO_2_ colloidal droplet is deposited onto the ellipsoidal structure from Figure 8b (prior to thermal treatment), it will not infiltrate the structure (depicted in Figure 9a). This is because all the pores are sealed due to the pre-self-assembled colloid. Consequently, the second droplet will form a torus shape above the structure.

This kind of superstructure can be seen very well on the photo-images performed during sample preparation. The structure takes a golden-brown color (Figure 9b) after thermal treatment and changes to gray with sparkling colored spots after silica sphere dissolution (Figure 9c). 

Optical microscopy images (Figure 9d–f) reveal striking and intense spots and geodesic lines generated by high-quality colloidal crystal reflections. The superstructure’s long-range order was confirmed through SEM investigations (Figure 9g,h) and UV–Vis spectroscopy (Figure 9h). Surprisingly, this spectrum closely resembles that of a “simple” ellipsoid superstructure (without the torus) from Figure 8i. To explore potential differences, a geometric optics investigation would be necessary, particularly regarding directional dependence.

A remarkable distinction emerges when the second silica colloidal drop from Figure 9a contains silica spheres of a different size (384 nm instead of 264 nm). In this scenario, the torus assumes a distinct lattice compared to the ellipsoid. Specifically, we observe three main reflection bands (Figure 9i):

λ_1_ = 703 nm: Attributed to (111) family planes provided by 384 nm holes in the torus, theoretically found at 715 nm.

λ_2_ = 590 nm: Associated solely with the (200) planes from the torus, featuring 384 nm holes. However, this band deviates significantly from its theoretical position (625 nm), possibly due to the torus surfaces’ poor quality.

λ_3_ = 483 nm: Attributed to (111) family planes provided by 264 nm holes in the ellipsoid, theoretically found at 491 nm.

While the presence of multiple bands confirms the overall good quality of the shaped superstructure, their deviations from theoretical positions highlight a significant contrast. Specifically, this contrast arises between two scenarios: (a)Inside infiltration (Figure 8f–g): In this case, the polystyrene reservoir is uniformly distributed within the silica crystalline structure volume.(b)Outside infiltration (Figure 9h,i): Here, the melted polystyrene must diffuse over a much longer distance (approximately 100 μm) through the silica crystalline structure. Notably, the tori in this scenario do not contain polystyrene spheres as reservoirs; instead, the polystyrene is provided from the ellipsoid.

However, researchers have already reported photonic hetero-structures with high-quality multiple reflection bands—narrow and distinct [74,75]. Interestingly, even more impressive examples exist in the natural world, specifically among diatoms [76,77,78], which possess frustules that closely resemble our colloidal superstructures. These remarkable living organisms exhibit light diffraction, iridescence, angle independence, and directional dependence—all achieved through their shaped superstructures [79,80,81,82]. Notably, some diatoms are comparable in size (around 1 mm) to our fabricated structures. The prospect of creating artificial superstructures with diatom-like light control efficiency is intriguing.

Furthermore, shaped colloidal photonic crystals do not require additional optical devices (such as geometric optics) when integrated into photonic platforms. For instance: a prismatic photonic crystal can replace a plane mirror, a concave photonic crystal can serve as a focusing lens, a convex photonic crystal can act as a dispersive lens, and an aspherical or freeform photonic crystal introduces additional degrees of freedom for manipulating light. We believe that the convergence of wave optics and geometric optics through the shape of photonic crystals could lead to significant advancements in photonics.

## 4. Conclusions

Shaped SiO_2_ colloidal crystals, which exhibit quadratic surfaces, can be self-assembled using the hanging-drop method with setups that deform the colloid drop shape. These crystals have macroscopic sizes ranging from 1 to 5 mm and a face-centered cubic (fcc) lattice ordered over long distances. Notably, they exhibit wide-angle independence and directional dependence in light diffraction, demonstrating shape-dependent variations in UV–Vis reflection bands when the working distance of the optical fiber probe is adjusted.

Shaped polystyrene superstructures, which exhibit quadratic surfaces, were fabricated as inverse-opal colloidal crystals. This was achieved by employing sequential self-assembly of spheres with different sizes and/or nature within a specialized architecture of a hanging drop, along with an inside infiltration approach. The resulting shaped superstructures—such as ellipsoids, caps of spheres, or tori on ellipsoids—present 3D meta-surfaces that strongly diffract and reflect light. These superstructures exhibit multiple reflection bands within the visible domain.

Our colloidal self-assembly occurs on a non-axisymmetric liquid/air interface of hanging drops by using setups that deform the shape of an ordinary spherical drop. The deformed drop surface acts as a topological interface, maintaining its shape over time and serving as a quality template for the self-assembly process.

## Figures and Tables

**Figure 1 polymers-16-01931-f001:**
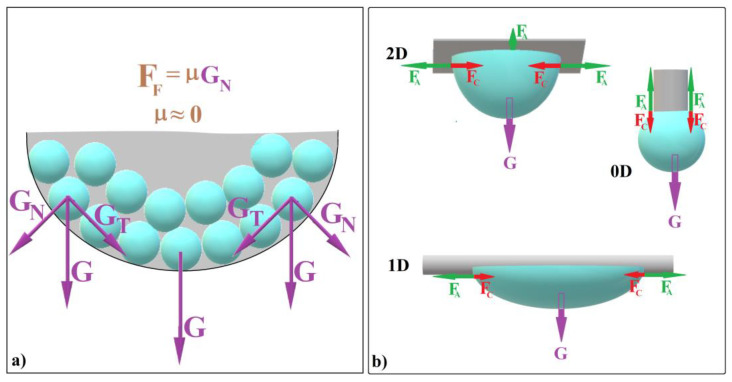
Schematic diagram of (**a**) colloidal self-assembly in a hanging drop and (**b**) equilibrium forces at the contact between a hanging drop and substrates with different dimensionalities.

**Figure 2 polymers-16-01931-f002:**
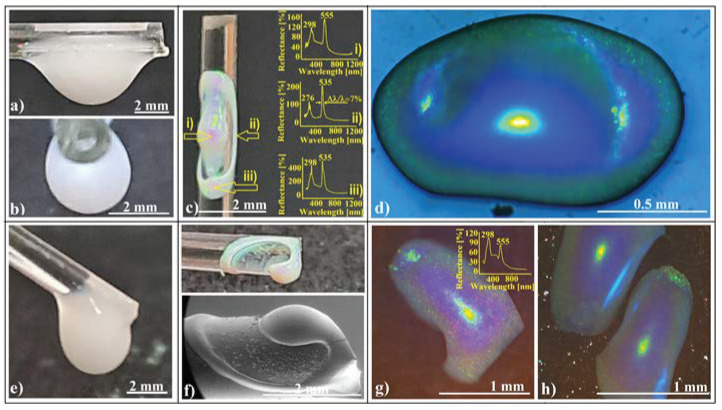
Photo-images of a silica sphere colloidal drop hanging on a horizontal glass fiber. (**a**) Front view; (**b**) end-on view; (**c**) photo-image of a colloidal crystalline structure self-assembled onto the horizontal glass fiber, inset: retro-reflection UV–Vis spectra from different corresponding zones; (**d**) optical microscopy image of an ellipsoidal colloidal photonic crystal self-assembled on a horizontal glass fiber; (**e**) photo-image of a colloidal drop hanging on a 45° tilted glass fiber, front view; (**f**) photo (upside) and SEM (downside) images of a colloidal crystalline structure self-assembled onto a 45° tilted glass fiber; (**g**,**h**) optical microscopy images of ellipsoidal colloidal photonic crystals self-assembled on a 45° tilted glass fiber, inset: retro-reflection UV–Vis spectrum on the ellipsoids.

**Figure 3 polymers-16-01931-f003:**
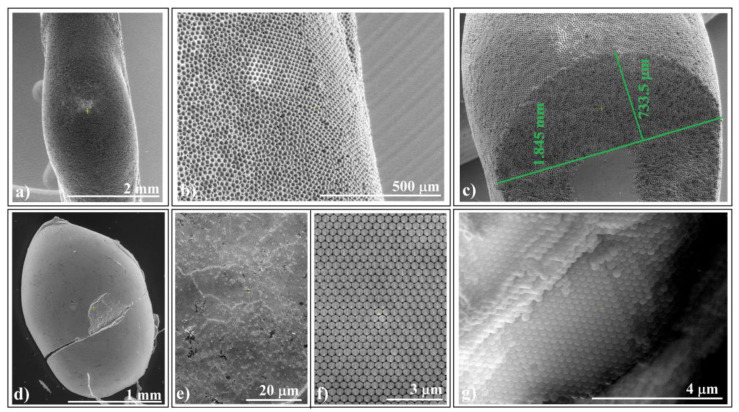
SEM images of ellipsoidal colloidal crystal self-assembled from: (**a**–**c**) 20 μm polystyrene spheres (far, surface, and cross-section view) and (**d**–**g**) 264 nm silica spheres (far, surface, close look, and cross-section view).

**Figure 4 polymers-16-01931-f004:**
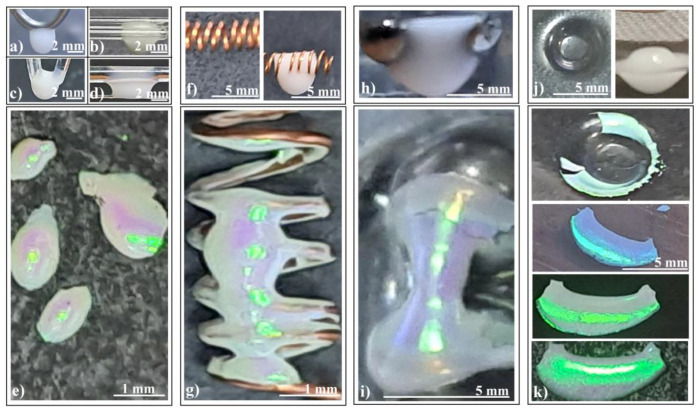
Photo-images of (**a**–**d**) silica sphere colloidal drops hanging on different fiber configurations, (**e**) ellipsoidal colloidal crystals self-assembled by different fiber configurations, (**f**) copper coil spring (**left**) and colloidal drop partially filling and hanging from the coil spring (**right**), (**g**) connected hyperboloids colloidal crystal self-assembled by coil spring, (**h**) colloidal drop hanging from two tangent metallic spheres, (**i**) hyperboloid colloidal crystal self-assembled by two tangent metallic spheres, (**j**) concavity formed in a polymeric sheet (**left**) and colloidal drop hanging from a concavity (**right**), and (**k**) cylindrical shell colloidal crystal self-assembled from a polymeric concavity, same fragment seen under different angles.

**Figure 5 polymers-16-01931-f005:**
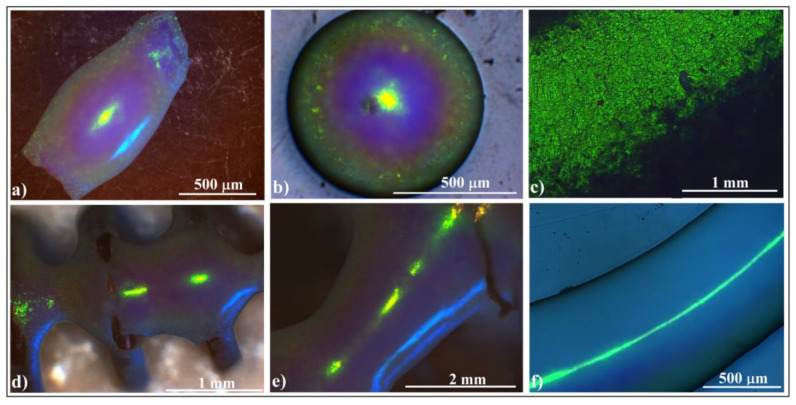
Optical microscopy images of colloidal photonic crystals with quadratic surfaces: (**a**) threeaxis ellipsoid, (**b**) cap of a sphere, (**c**) cylindrical shell, (**d**) connected hyperboloids, (**e**) hyperboloid, and (**f**) torus.

**Figure 6 polymers-16-01931-f006:**
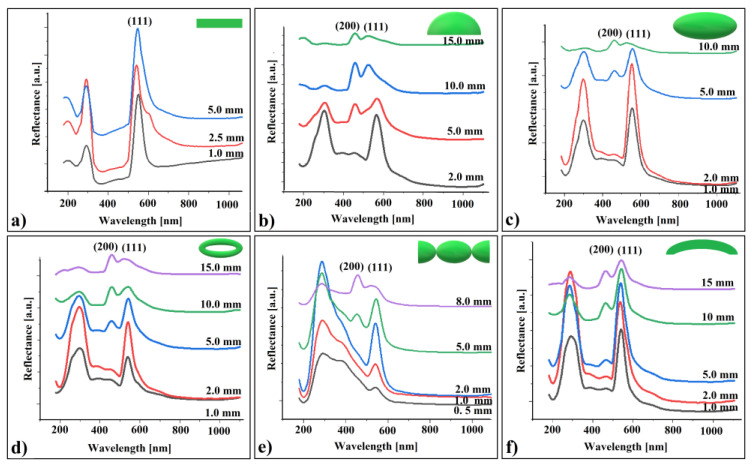
UV–Vis retro-reflection spectra of shaped colloidal crystals from different working distance: (**a**) flat, (**b**) cap of a sphere, (**c**) ellipsoid, (**d**) torus, (**e**) connected hyperboloids, and (**f**) cylindrical shell.

**Figure 7 polymers-16-01931-f007:**
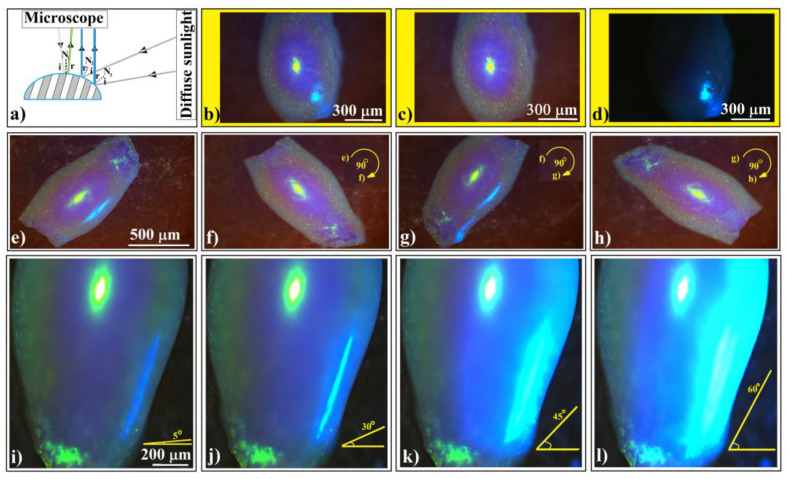
(**a**) Schematic diagram of experimental setup in which an ellipsoidal colloidal crystal is simultaneously or alternatingly illuminated by two white light sources. Optical microscopy images of an ellipsoidal colloidal photonic crystal illuminated by (**b**) both normal microscope and diffuse sided white light sources, (**c**) only normal microscope light, and (**d**) only diffuse sided light. (**e**–**h**) Optical microscopy images of an ellipsoidal colloidal photonic crystal illuminated by both light sources, but the sample is in-plane rotated by 90°. (**i**–**l**) Optical microscopy images of an ellipsoidal colloidal photonic crystal illuminated by both light sources, while keeping constant the normal microscope light but increasing the angular illumination from the side source.

**Figure 8 polymers-16-01931-f008:**
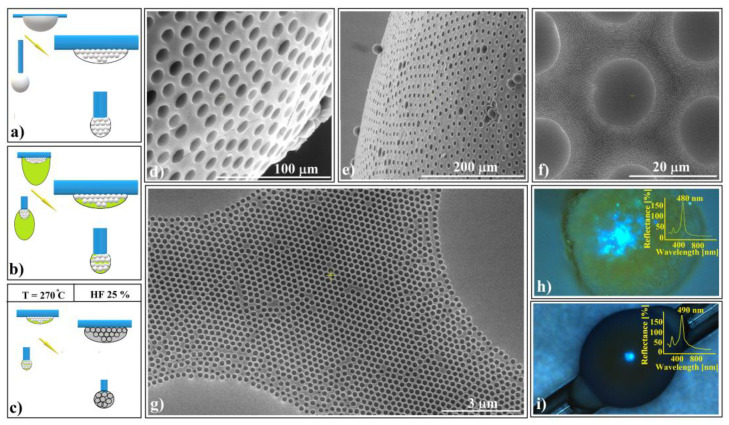
Schematic diagram of (**a**) 20 μm polystyrene spherical and ellipsoidal colloidal crystal self-assembly on a horizontal and vertical fiber, respectively, (**b**) 264 nm SiO_2_ colloid infiltration in 20 μm polystyrene ellipsoidal colloidal crystals, (**c**) thermal and SiO_2_ hydrofluoric acid dissolution steps, necessary for shaped super-structured inverted opal fabrication; SEM images of (**d**) spherical- and (**e**) ellipsoidal-shaped super-structured inverted opals, (**f**,**g**) shaped super-structured inverted opals’ surface structure; optical microscopy images of shaped super-structured inverted opals: (**h**) sphere (inset—UV–Vis retro-reflection spectrum) and (**i**) ellipsoid (inset—UV–Vis retro-reflection spectrum).

**Figure 9 polymers-16-01931-f009:**
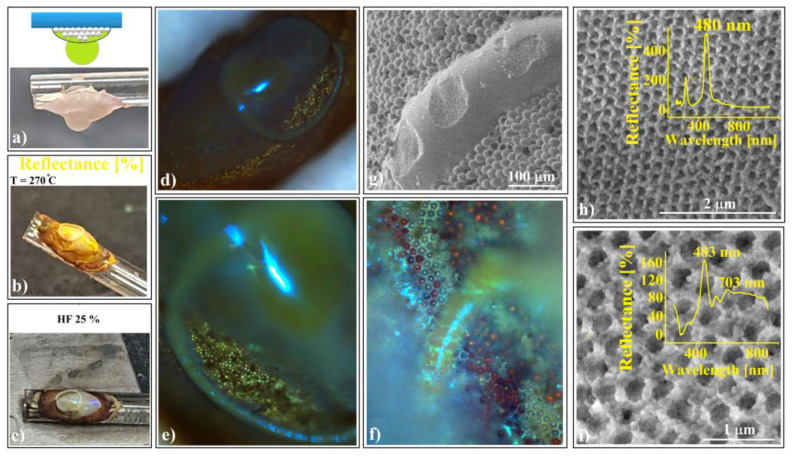
Torus onto ellipsoid inverse-opal-shaped superstructure; (**a**) the second SiO_2_ spheres colloidal drop hanging on a PS/SiO_2_ ellipsoid, (**b**) after thermal treatment, and (**c**) after HF dissolution; (**d**–**f**) optical microscopy images of torus onto ellipsoid; (**g**) SEM image of torus onto ellipsoid; (**h**,**i**) SEM images of torus surfaces with different hole sizes (inset—corresponding UV–Vis spectroscopy spectra, (**h**) 264 nm and (**i**) 384 nm).

## Data Availability

Data are contained within the article.

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
