# Peer review of "Shaping in the Third Direction: Colloidal Photonic Crystals with Quadratic Surfaces Self-Assembled by Hanging-Drop Method"

_polymers, 2024, doi:10.3390/polym16131931_

Round 1

Reviewer 1 Report

Comments and Suggestions for Authors

Comments for the authors are submitted as a separate pdf document to the system.

Comments on the Quality of English Language

A thorough proofreading by a native English speaker or a professional editor is recommended to catch any minor grammatical errors and improve the overall fluency of the text.

Author Response

The paper, titled "Shaping in the Third Direction: Colloidal Photonic Crystals with Quadratic Surfaces Self-Assembled by Hanging Drop Method" by Ion Sandu et al., investigates the fabrication of 3D-shaped colloidal photonic crystals. Utilizing silicon dioxide and polystyrene spheres with diameters of 0.264 µm and 20.00 µm respectively, the research focuses on assembling these materials into complex shapes such as ellipsoids and hyperboloids through a hanging drop method. The study highlights how these shapes can influence optical properties due to their geometric configurations, potentially enhancing applications in optical devices and spectrophotometry. The optical phenomena are quantitatively analyzed, including the photonic band gaps and reflection properties of the materials, underlining the potential for new types of aspheric and freeform diffracting mirrors and metasurfaces.

Major Issues:

  1. Reviewer: The novelty of using the hanging drop method to create non-spherical colloidal photonic crystals is well demonstrated. However, the paper could benefit from a deeper comparison with existing methods of fabricating 3D photonic crystals. Please include a detailed comparison of the hanging drop method with other self-assembly and fabrication techniques, highlighting the advantages and limitations of each.

 Authors:  A deeper and detailed comparison between all the methods which are able to

 produce a photonic crystal, or a colloidal crystal, including the hanging drop method

              could be an interesting idea for a Review article. Unfortunately, not only that the

              allocated space for the article Introduction part would be insufficient, but the other

              Reviewers assigned for this Manuscript suggested a drastic shortening of the original

              Introduction. However, the present Manuscript contains in figure 1 the basic principles

              and features of hanging drop method. Moreover, the actualized references [61] and [39]

              contain a partial answer to your request. Shortly, it would be great but seems to be

              impossible.

  1. Reviewer: The optical characterization of the colloidal photonic crystals is thorough, but the impact of different environmental conditions (e.g., temperature, humidity) on the optical properties is not discussed. Discuss the stability of the optical properties under varying environmental conditions. Consider including experimental data on the effects of temperature and humidity on the photonic band-gap properties.

 Authors: If the influence of temperature or humidity on the morphological and structural   characteristics of the shaped colloidal crystals self-assembled by hanging drop method were mentioned in this article, the influence of varying environmental conditions on the optical properties of shaped colloidal crystals is a huge task which requires hundreds of particular experiments. However, we have no reason to believe that some experiments performed in the actualized reference [39] (bandgap variation with glucose concentration in aqueous media filling the pores) would give different results on the present ellipsoidal colloidal crystals. However, only future experiments could prove or invalidate this presumption.

  1. Reviewer: The paper outlines potential applications but lacks detailed examples of practical implementations. Please provide more specific examples of potential applications for these shaped photonic crystals, such as in optical sensors, waveguides, or lenses. Discuss how these crystals could be integrated into existing technologies.

 Authors: Optical computers components, spectrometers, freeform diffractive-reflexive optical couplers.  Even if we may correctly identify the function of an existing optical device with an optical ability of our shaped colloidal crystal, auxiliary devices must be architectured and fabricated. This will take some time and efforts.

Minor Issues:

  1. Reviewer: The methodology section is comprehensive, but certain experimental details are missing. It would be beneficial for readers if the authors can provide more detailed information on the preparation and handling of colloidal solutions, including specific concentrations, drying times, and temperatures used in the self-assembly process.

Authors: All of these data were presented. Any other information are not necessary.

  1. Reviewer: The figures are clear but could be more informative. Including more annotated diagrams to explain the self-assembly process and the formation of different shapes can positively impact the paper. High-resolution images showing the defects or imperfections in the crystals would also be beneficial.

Authors:  Figure 1 contains the main characteristics of the hanging drop method. A new subfigure was added to figure 7 (the 7a scheme)The main imperfection of the classical flat colloidal crystals are their cracks. It can be seen in SEM images from figure 3 the lack of cracks in the case of our curved colloidal crystals.

  1. Reviewer: The introduction covers relevant background information, but there are some recent advancements in the field that are not cited. Update the literature review to include recent studies on 3D photonic crystals and self-assembly techniques from the past three years. This will provide a more current context for the research.

Authors:  Even we scan few hundreds of scientific research articles concerning self -assembly and 3D photonic crystals we were forced to restrict their number that could be cited to only those that were useful. However, their number in our References list is of 16 that were published between 2022 and 2024 which we consider as a reasonable number.

Other Concerns:

Oth-1. What parameters (e.g., drop volume, substrate tilt angle) were found to be most critical in achieving high-quality self-assembly?

Authors: This may be one of the special features of the hanging drop when compared to other self-assembly methods; their high-quality self-assembly is always the same, no matter that a huge number of parameters (external or internal) were varied in experiments. However, as any other self-assembly method is highly sensitive to the size dispersion of the spheres from colloidal solution.

Oth-2. How reproducible is the hanging drop method across different experimental setups?

Authors: For the reason from Oth-1, the method is highly reproducible. Text was added in Experimental paragraph:

“2.3 Reproducibility

The hanging drop method is a highly reproducible one. The specific shape of colloidal crystals is obtained each time within the limits of experimental parameters. The high quality of spheres ordering is not influenced by synthesis parameters such as: relative humidity, temperature (up to 80 °C), concentration (up to 10 wt.%), external vibrations and shocks, or their short transitory variations.  However, as any other self-assembly method, is highly sensitive to size dispersion of the spheres from colloidal solution.”

Oth-3. How do the optical properties vary with changes in the size and shape of the colloidal particles?

Authors: The optical properties of our curved colloidal crystals composed of spherical units are generated by geometric optics while the wave phenomena are the same as those on flat colloidal crystals. Changing the spherical units with other shaped units would not change significant the geometrical phenomena. However, experiments must be done.

Oth-4. Are there specific shapes that exhibit superior optical performance for certain applications?

Authors:  The ellipsoids seem to have interesting potential applications in which their specific iridescence would allow fabrication of fixed optical devices (telescope lenses). Also, their enhanced field of view could be useful in hard to point applications (maybe telescope mirrors).

Oth-5. What are the challenges in scaling up the hanging drop method for industrial applications?

Authors:  The method is so simple, thus it could be easily scaled-up by using only robotic systems.

Oth-6. Have any strategies been identified to overcome these challenges?

Authors:  We identified no challenges.

Oth-7. What types of defects are most commonly observed in the self-assembled structures?

Authors:  Mostly, stacking faults.

Oth-8. How do these defects impact the overall optical performance of the crystals?

Authors:  They enlarge the reflections bands (larger FWHM).

Oth-9. What are the practical considerations for integrating these shaped photonic crystals into existing optical devices?

Authors:  Even if we may correctly identify the function of an existing optical device with an optical ability of our shaped colloidal crystal, auxiliary devices must be architectured and fabricated. This will take some time and efforts

Reviewer 2 Report

Comments and Suggestions for Authors

This work is devoted to the fabrication of photonic crystals based on SiO2 nanoparticles. Also authors invesitigated the influence of a 20 um polystyrene spheres on photonic crystal formation. Please, specify novelty of the research. What is the influence of temperature and concentration of SiO2 on the photonic crystal formation? What was zeta-potential of SiO2 spheres? Panels on Fig. 7 looks too similar, please, provide scheme which identidy difference or shorten amount of panels. For all figures: please, increas font of panel numbering (a),b), etc), thicnkness of scale bars and font in insets. Overall, this paper is worth to be published after addressing of listed issues.

Comments on the Quality of English Language

Minor editing is required

Author Response

This work is devoted to the fabrication of photonic crystals based on SiO2 nanoparticles. Also authors invesitigated the influence of a 20 um polystyrene spheres on photonic crystal formation. Please, specify novelty of the research. What is the influence of temperature and concentration of SiO2 on the photonic crystal formation? What was zeta-potential of SiO2 spheres? Panels on Fig. 7 looks too similar, please, provide scheme which identidy difference or shorten amount of panels. For all figures: please, increase font of panel numbering (a),b), etc), thicnkness of scale bars and font in insets. Overall, this paper is worth to be published after addressing of listed issues.

  1. Reviewer: Please, specify novelty of the research.

    Authors: Text was added to the final of Introduction. “The aim of this work is to find the experimental conditions in which colloidal particles self-assemble as quality colloidal photonic crystals, other than spherical ones that were obtained in ref. [39]. We report as a novelty the synthesis of freeform colloidal photonic crystals by self-assembly method and some of their shape dependent optical phenomena.”

  1. Reviewer: What is the influence of temperature and concentration of SiO2 on the photonic crystal formation?

    Authors: Text was added before subchapter 3.2:  “The hanging drop method is weak dependent on relative humidity in the air, rate of evaporation, temperature or concentration. However, the colloidal crystal shape and its mechanical resistance are assured by a large amount of solid spheres which usually is provided by a high colloid concentration, while their organization quality by a limited concentration (up to 10 % wt.). These opposite requirements can be accommodate by using large volumes of initial colloidal solutions in contact with the substrates. This means a limited range of substrates which can assure a high adherence force or/and an increased contact surface between substrates and liquid colloids.”

  1. Reviewer: What was zeta-potential of SiO2 spheres?

    Authors: Our zeta-potential of SiO2 spheres is unknown since the producer (MicroParticles gmbh) did not specified it neither in product datasheet nor their Web Site. However, one product datasheet placed on Internet by an anonymous source shows a value of Z = -48 mV for SiO2 (50 nm) water based colloidal spheres produced by ‘’MicroParticles gmbh”.

  1. Reviewer: Panels on Fig. 7 looks too similar, please, provide scheme which identify difference or shorten amount of panels.

    Authors: Schematic was introduced in Fig. 7

  1. Reviewer: For all figures: please, increase font of panel numbering (a),b), etc), thicnkness of scale bars and font in insets.

Authors: For all figures and insets, font was increased. Also scale bars thickness was increased.

Reviewer 3 Report

Comments and Suggestions for Authors

The manuscript shows results of original studies of a method which the authors work on, as a continuation on refs. 40 and 72. The method of opals fabrication seems interesting, because gravitational force helps in self-organization of the composing nanospheres and in my opinion works such like this brings novel viewpoint on fabrication of photonic crystals with curved surfaces. This manuscript extends the research of recently published results in ref. 40. Some similiarities of the experimental details is found, however this reseach deals more on different shapes of colloidal crystals obtained by different shapes of the 'substrates' and characterization of optical properties on curved surfaces of ellopsoidal photonic crystals. 

In my opinion the introduction is far too large and should be shortened greatly (2-3 times). In fact, it contains very interesting information in general, but should be reedited towards precisely this article. Broad literature study should be dedicated more to the review articles.  Here , the introduction should describe concisely the background of the research. Individual paragraphs should refer strictly to the research topic. or the relationship should be more clearly defined.

Maybe, much of the information may be used further in the discussion, instead? 

l131-135: different criteria are for self-assembly processes. What is the purpose to have them listed in one place?

moreover, the information is so general. why starting from the beginning at this place?

l154-167: not clear description. Points a-d does not describe strictly "the role that [...] water plays [...]" , but e.g. of capillary forces, colloidal particles. I would avoid above statement. 

In point c) "particles [...] move to the substrate", then in point d) they "[...] move to the liquid/air interface". Explanation with this respect should be given.

In the introduction it should be more clearly stated, the research novelty with respect to the ref. [40]

Fig.1a:

the forming vector (GN and GT) with their actual length do not sum up geometrically to G

Fig. 1b

The representation of the idea is not clear.

The vectors FC, FA and G do not have an anchor point on any object.

It is strongly recommended that the last paragraph of the Introduction shoul describe aim of the work and methodology.

"Alveoli" is a professional name in the packaging? Could not it be e.g. (air) bubbles ?

2.2.1.1 - what size of droplets or were formed typically?

4-number numeration in subparagraphs of 2.2.1. and 2.2.2 should be ommited .

Numeration of sections after "3.Results and Discussion" is wrong

Generally , the points of each of the Figures a), b), c),... are too small and hard to read. They should be enlarged

l322-326: The grating effect is rather hard to observe on the systems with periodicity of around 264 nm. The diffraction angle for such periodicity would be very high. I suppose , that different colors of reflection are rather because of Bragg reflection on a curved surface, producing different reflection wavelength.  

It would be interensting to comment on how different type of surface materials (e.g. metals, plastics) affect the shape of the droplets. How does the surface tension or contact angle of water droplets changes (even theoretically)?

l476: what means "windows only" ? it sounds like jargon. in my opinion should be ommited . (and l507)

page 13: A scheme with an ellipsoid and/or light incident on the object might be helpful to depict some setup in discussion

 l592-593: "geodesic lines" - confusing statement

Comments on the Quality of English Language

Extensive english corrections should be made (even in the abstract): mistakes, unclear or confusing sentences. Some should be stated more precisely/clearly. Below are listed some of them:

lines 11-12: "wide angle independent" , "reflective diffraction crystals", between 1-5 mm

l14-15: highly similar with ... 

"tor"

l21: Such as 3D shaped ...

l181: "bumped colloidal crystal"

l189-190: "shape as time as its bottom"

l255: "fails to infiltrate" 

l275: "An almost spherical one"

l285 & l288-289: "self-assembled onto"

l418: A flat colloidal crystal capillary self-assembled between two parallel glass slides

l423, l 431: cape of sphere

l431: tor

Author Response

The manuscript shows results of original studies of a method which the authors work on, as a continuation on refs. 40 and 72. The method of opals fabrication seems interesting, because gravitational force helps in self-organization of the composing nanospheres and in my opinion works such like this brings novel viewpoint on fabrication of photonic crystals with curved surfaces. This manuscript extends the research of recently published results in ref. 40. Some similiarities of the experimental details is found, however this reseach deals more on different shapes of colloidal crystals obtained by different shapes of the 'substrates' and characterization of optical properties on curved surfaces of ellopsoidal photonic crystals.

  1. Reviewer: In my opinion the introduction is far too large and should be shortened greatly (2-3 times). In fact, it contains very interesting information in general, but should be reedited towards precisely this article. Broad literature study should be dedicated more to the review articles. Here, the introduction should describe concisely the background of the research. Individual paragraphs should refer strictly to the research topic. or the relationship should be more clearly defined. Maybe, much of the information may be used further in the discussion, instead?

 Authors: The Introduction part was reduced to the half. A part of it was moved in Results and Discussion and another part was completely eliminated.

  1. Reviewer: 131-135: different criteria are for self-assembly processes. What is the purpose to have them listed in one place? moreover, the information is so general. why starting from the beginning at this place?

 Authors: This part was completely eliminated.

  1. Reviewer: l154-167: not clear description. Points a-d does not describe strictly "the role that [...] water plays [...]", but e.g. of capillary forces, colloidal particles. I would avoid above statement.

 Authors: This part was completely eliminated.

  1. Reviewer: In point c) "particles [...] move to the substrate", then in point d) they "[...] move to the liquid/air interface". Explanation with this respect should be given.

 Authors: This part was completely eliminated.                

  1. Reviewer: In the introduction it should be more clearly stated, the research novelty with respect to the ref. [40]

Authors: Text was added at the end of Introduction Chapter: “In this work we report for the first time the freeform colloidal photonic crystals synthesis by self-assembly method and some of their shape dependent optical phenomena. “

  1. Reviewer Fig.1a: the forming vector (GN and GT) with their actual length do not sum up geometrically to G

Authors: Fig. 1a was modified.

  1. Reviewer Fig. 1b The representation of the idea is not clear. The vectors FC, FA and G do not have an anchor point on any object.

Authors: Fig. 1b was modified.

  1. Reviewer It is strongly recommended that the last paragraph of the Introduction should describe aim of the work and methodology.

Authors: Text was added to the final of Introduction: “The aim of this work is to find the experimental conditions in which colloidal particles self-assemble as quality colloidal photonic crystals, other than spherical ones that were obtained in ref. [39]. We report for the first time the synthesis of freeform colloidal photonic crystals by self-assembly method and some of their shape dependent optical phenomena.”

  1. Reviewer "Alveoli" is a professional name in the packaging? Could not it be e.g. (air) bubbles?

Authors: The terms alveolus and alveoli were replaced with concavity and concavities, respectively in the manuscript text and fig.4 legend.

  1. Reviewer 2.2.1.1 - what size of droplets or were formed typically?

Authors: Text was added (before the fig.2): “Using a large volume (tens of microliters, 3 - 5 mm in size) of 264 nm SiO2 colloidal solution (Figure 2a, b)”

  1. Reviewer 4-number numeration in subparagraphs of 2.2.1. and 2.2.2 should be ommited. Numeration of sections after "3.Results and Discussion" is wrong

Authors: Changes were performed.

  1. Reviewer Generally, the points of each of the Figures a), b), c),... are too small and hard to read. They should be enlarged

Authors: For all figures and insets, the font was increased. Also scale bars thickness was increased.

  1. Reviewer l322-326: The grating effect is rather hard to observe on the systems with periodicity of around 264 nm. The diffraction angle for such periodicity would be very high. I suppose , that different colors of reflection are rather because of Bragg reflection on a curved surface, producing different reflection wavelength.

Authors: Another hypothesis is proposed. Text was added (after phrase where fig.2d is described): “The iridescence phenomenon (green to blue transition) is attributed to Bragg reflection on a convex surface (mirror) which presents a higher field of view and a higher angular field of view than a planar one. Bragg diffracted light at large incidence angles will be reflected inside the acceptance cone of the microscope objective while in the case of a plane mirror it will fall outside.”

  1. Reviewer It would be interesting to comment on how different type of surface materials (e.g. metals, plastics) affect the shape of the droplets. How does the surface tension or contact angle of water droplets changes (even theoretically)?

Authors: Text was added: “By using as substrates fibers of different nature (glass, metal, plastic) which present different contact angle at their triple contact line with the colloidal drop we observed that lower contact angle such as glass/water one supports a higher volume of colloid against gravity than metal or plastic. This leads to thicker and more mechanical resistant colloidal crystals. By mixing ethanol with colloidal solution in different ratios, the adherence force between colloid and fiber increases, elongating the drops along the fiber. However, this leads also to the decreases of cohesion force, thus a much lower volume can be suspended and subsequently, thinner and more elongated colloidal crystals results.”

  1. Reviewer l476: what means "windows only" ? it sounds like jargon. in my opinion should be ommited . (and l507)

Authors: 476 Line phrase was clarified as “which depicts three distinct situations: both light sources, microscope light only, and diffuse sunlight (coming in through the window) only.” 507 Line phrase was also modified: “Simultaneously, we adjusted the maximum angle at which light from the secondary source (the diffuse sunlight coming in through the window) fell on the crystal’s side surface - a kind of solid angle.”

  1. Reviewer page 13: A scheme with an ellipsoid and/or light incident on the object might be helpful to depict some setup in discussion

Authors: A schematic was introduced in Fig. 7 as new subfigure 7a, as well as their description in the fig 7a legend :” a) Schematic of experimental set-up in which an ellipsoidal colloidal crystal is simultaneous or alternative illuminated by two white light sources”; the letters for the rest of subfigures were renamed accordingly in fig.7 legend and manuscript text.

  1. Reviewer l592-593: "geodesic lines" - confusing statement

Authors: Usually, the geodesic lines are defined as shortest paths between two points on a curved surface. The l592-593 phrase was corrected as:  “Optical microscopy images (Figure 9d-f) reveal striking and intense spots along certain geodesic lines generated by high-quality colloidal crystal reflections.

  1. Reviewer Extensive english corrections should be made (even in the abstract): mistakes, unclear or confusing sentences. Some should be stated more precisely/clearly. Below are listed some of them:

lines 11-12: "wide angle independent" , "reflective diffraction crystals", between 1-5 mm

l14-15: highly similar with ...

"tor"

l21: Such as 3D shaped ...

l181: "bumped colloidal crystal"

l189-190: "shape as time as its bottom"

l255: "fails to infiltrate"

l275: "An almost spherical one"

l285 & l288-289: "self-assembled onto"

l418: A flat colloidal crystal capillary self-assembled between two parallel glass slides

l423, l 431: cape of sphere

l431: tor

Authors: Suggested corrections were made in the manuscript text and marked with red letters

"wide angle independent" corrected as "wide-angle independent”

"reflective diffraction crystals" corrected as " diffractive reflection crystals"

"between 1-5 mm" corrected as " between 1 and 5 mm"

"highly similar with ... " corrected as " highly similar to ... "

"tor" corrected as "torus"

"Such as 3D shaped ... " corrected as "Such as 3D-shaped ... " as in scientific literature

"bumped colloidal crystal" corrected as "bump-like colloidal crystal"

“also independent of the droplet’s shape as time as its bottom remains contact-free” was corrected as

“also independent of the droplet’s shape as time as its lower part remains contact-free”

“ensuring that the liquid/air interface at the bottom remains contact-free“ was corrected as “ensuring that the liquid/air interface at the lower part remains contact-free“

“The second drop fails to infiltrate (all holes are full)” was corrected and clarified as “The second drop can not infiltrate (all former holes between PS spheres are already full with silica spheres)”

"An almost spherical one" corrected as "A spheroidal one"

"self-assembled onto"   corrected as "self-assembled on"

"A flat colloidal crystal capillary self-assembled between two parallel glass slides"   corrected as "A flat colloidal crystal self-assembled by capillarity between two parallel glass slides"

"cape of sphere" corrected as "cap of sphere"

Also, the list of references and their corresponding  numbers were modified (if required)  according with the modifications in the manuscript

Round 2

Reviewer 1 Report

Comments and Suggestions for Authors

The comments provided by the authors are reasonable, and I find them satisfactory. However, I have some concerns about the originality of the study and have sent my detailed comments to the Editor.

Comments on the Quality of English Language

Language of the manuscript is fine.

Author Response

Answer to reviewers

Reviewer 1. Dear Authors, Please find enclosed the Reviewer's comments on your article. As you might notice there are still some concerns about the similarity of this work as compared with your previously published work in Polymers. We can consider this article to be published after proper reply to the Reviewer concerns. Please try to highlight any major differences between the two works. Best regards,

Authors: 
The main idea and novelty of our previously published work in Polymers, “Shaping in the Third Direction: Self-Assembly of Convex Colloidal Photonic Crystals on an Optical Fiber Tip by Hanging Drop Method. Polymers 2024, 16, 33. https://doi.org/10.3390/polym16010033”, is the synthesis of a high quality colloidal photonic crystal on the tip of an optical fiber. The macroscopic shape of the crystal was hemispherical and it can be achieved only if the fiber is vertically oriented.  It is not a trivial idea to think that by changing the fiber orientation and the place of the drop on the fiber you may obtain colloidal crystal of a macroscopic shape (ellipsoidal) different than the hemispherical one and even more that the colloidal crystal will preserve their long range order.  
If in our previously published work a single kind of substrate was used (a 400 mm optical fiber core), in the present work we used single or multiple different fibers: simple glass, metallic or polymer forming different architectures, or other 3D shaped substrates such as: metal springs, metal spheres, or polymer concavities which have as result colloidal photonic crystals macroscopic shaped like a hyperboloid or connected hyperboloids, or Pascal snail. It is hard to believe that one can obtain different things doing same activities.  
A secondary objective of the previously published work in Polymers was to use the hemispherical colloidal crystals as opto-chemical sensors. The present article has as a secondary objective to highlight geometric optic effects such as direction dependency of reflected light, or multiple optical band-gaps revealed by angular field of view variation or by using super structured inverse opals like.  
In the present article we cited (as ref. 39) several times the previously work published in Polymers, such as: 
“Curved colloidal crystals - such as spheres, hemispheres, cylinders, doughnut-like shapes, or tori - have emerged through self-assembly processes [31-40]. “ were we mentioned that by hanging drop method we obtained “Curved colloidal crystals”. However, curved can comprise a high number of different objects. Hemispherical curved is quite different than ellipsoidal curved!
“However, we have encountered a small number of suggested applications related to shape dependence different from that of wavelength angular independence [39, 40]”. These applications leverage simultaneous Bragg and grating diffraction phenomena in convex colloidal crystals.” It refers to the property of a curved colloidal crystal to keep its color when changing the incidence and reflected angle, in opposition with a planar colloidal crystal. However, hemispherical, ellipsoidal or hyperboloid shaped crystals are all convex. In the previously article we spotted this phenomena only for the hemispherical one. 
“The aim of this work is to find the experimental conditions in which colloidal particles self-assemble as quality colloidal photonic crystals, other than spherical ones that were obtained in ref. [39]”. We think that we expressed clearly: “other than spherical ones”. This is the major difference. And we must to mention again that is not so obvious how to proceed to obtain free-form colloidal photonic crystals. In the present article we shown only a small number of our experiments, presenting only the successful ones.

In the aim of clarifying the differences between the approaches used in previously article and present work, text was added: 
lines 222-224, “As the drop dries (Fig. 1b, case 2D and case 0D [39]), it forms a spherical colloidal crystal. If the drop hangs on a 2D substrate it always forms a spheroidal colloidal crystal. If a drop hangs on a 0D substrate it always forms a cap of sphere colloidal crystal.
lines 235-236, “Would this lead to the self-assembly of an elongated colloidal crystal ?”
lines 304-306. “When a colloidal drop hangs on a tilted fiber (at an angle of 30°-60°), the resulting crystalline structure becomes even more captivating, quite different from both those resulted when a drop is placed on a vertically fiber tip [39] or when the drop hangs on a horizontally fiber (Figure 2c).”

Reviewer 2 Report

Comments and Suggestions for Authors

Authors addressed all listed issues. Paper can ba published in present form

Comments on the Quality of English Language

Minor editing of English language required

Author Response

Once the manuscript is approved for publication, we will apply for the English editing professional service

Round 3

Reviewer 1 Report

Comments and Suggestions for Authors

I have no other objections or comments regarding the revised study in its current version. I recommend publishing the study after a proper check for English.

Comments on the Quality of English Language

It is recommended to have the English language checked by a native speaker.

Author Response

Reviewer 1. Dear Authors, Please find enclosed the Reviewer's comments on your article. As you might notice there are still some concerns about the similarity of this work as compared with your previously published work in Polymers. We can consider this article to be published after proper reply to the Reviewer concerns. Please try to highlight any major differences between the two works. Best regards,

Authors:

  1. The main idea and novelty of our previously published work in Polymers, “Shaping in the Third Direction: Self-Assembly of Convex Colloidal Photonic Crystals on an Optical Fiber Tip by Hanging Drop Method. Polymers 2024, 16, 33. https://doi.org/10.3390/polym16010033”, is the synthesis of a high quality colloidal photonic crystal on the tip of an optical fiber. The macroscopic shape of the crystal was hemispherical and it can be achieved only if the fiber is vertically It is not a trivial idea to think that by changing the fiber orientation and the place of the drop on the fiber you may obtain colloidal crystal of a macroscopic shape (ellipsoidal) different than the hemispherical one and even more that the colloidal crystal will preserve their long range order.  
  2. If in our previously published work a single kind of substrate was used (a 400 mm optical fiber core), in the present work we used single or multiple different fibers: simple glass, metallic or polymer forming different architectures, or other 3D shaped substrates such as: metal springs, metal spheres, or polymer concavities which have as result colloidal photonic crystals macroscopic shaped like a hyperboloid or connected hyperboloids, or Pascal snail. It is hard to believe that one can obtain different things doing same activities.
  3. A secondary objective of the previously published work in Polymers was to use the hemispherical colloidal crystals as opto-chemical sensors. The present article has as a secondary objective to highlight geometric optic effects such as direction dependency of reflected light, or multiple optical band-gaps revealed by angular field of view variation or by using super structured inverse opals like.
  4. In the present article we cited (as ref. 39) several times the previously work published in Polymers, such as:
  5. “Curved colloidal crystals - such as spheres, hemispheres, cylinders, doughnut-like shapes, or tori - have emerged through self-assembly processes [31-40]. “ were we mentioned that by hanging drop method we obtained “Curved colloidal crystals”. However, curved can comprise a high number of different objects. Hemispherical curved is quite different than ellipsoidal curved!
  6. “However, we have encountered a small number of suggested applications related to shape dependence different from that of wavelength angular independence [39, 40]”. These applications leverage simultaneous Bragg and grating diffraction phenomena in convex colloidal crystals.” It refers to the property of a curved colloidal crystal to keep its color when changing the incidence and reflected angle, in opposition with a planar colloidal crystal. However, hemispherical, ellipsoidal or hyperboloid shaped crystals are all convex. In the previously article we spotted this phenomena only for the hemispherical one.
  7. “The aim of this work is to find the experimental conditions in which colloidal particles self-assemble as quality colloidal photonic crystals, other than spherical ones that were obtained in ref. [39]”. We think that we expressed clearly: “other than spherical ones”. This is the major difference. And we must to mention again that is not so obvious how to proceed to obtain free-form colloidal photonic crystals. In the present article we shown only a small number of our experiments, presenting only the successful ones.

In the aim of clarifying the differences between the approaches used in previously article and present work, text was added: lines 222-224, lines 235-236, lines 304-306.
